# Use of the Health Improvement Card by Chinese physical therapy students: A pilot study

Xubo Wu[1,2☉], Alice YM Jones[3,4‡]*, Yiwen Bai[1,2☉], Jia Han[5,6,7☉], Elizabeth Dean[8‡]

**1** Shanghai University of Traditional Chinese Medicine, Department of Physical Therapy, Shanghai, China, **2** Seventh People's Hospital of Shanghai University of Traditional Chinese Medicine, Department of Rehabilitation Medicine, Shanghai, China, **3** The University of Sydney, Faculty of Health Sciences, Sydney, NSW, Australia, **4** The University of Queensland, School of Health and Rehabilitation Sciences, Brisbane, QLD, Australia, **5** Shanghai University of Sports, Physiotherapy and Sports Rehabilitation Department, Shanghai, China, **6** University of Canberra, Research Institute for Sport and Exercise, Canberra, ACT, Australia, **7** Swinburne University of Technology, Faculty of Health, Arts and Design, Melbourne, VIC, Australia, **8** University of British Columbia, Department of Physical Therapy, Vancouver, Canada

☉ These authors contributed equally to this work.
‡ These authors also contributed equally to this work.
* alice.jones@sydney.edu.au

**Data Availability Statement:** All relevant data are within the manuscript and its Supporting Information files.

**Funding:** This work was supported by Shanghai Municipal Health Bureau, project ZY (2018-2020) -

## Abstract

This study investigated the perceptions of Chinese physical therapy students on use of the Health Improvement Card (HIC) as a clinical tool to assess lifestyle and prescribe health education to others. The biometrics and health indices/attributes/lifestyles of these students were also evaluated with self-administration of the HIC. After a tutorial on the HIC and its clinical application, physical therapy students (n = 82) from two Chinese universities, completed the Chinese translation of the HIC followed by a questionnaire on students' perceptions of it. Second, they invited a friend/relative to complete the HIC. Then, they provided feedback on the HIC's strengths and challenges related to its administration. The data were analyzed with descriptive statistics and content thematic analysis. Response rate of self-completed HICs was 100% (n = 82) and that of questionnaires was 99% (n = 81). Participants' age range was 20–34 years; mean body mass index (BMI) was 23.9±5.4 for men and 20.5±2.6 kg/m$^2$ for women. Generally, participants had low-risk BMIs (82%) and blood pressures (BPs) (91%), moderate-risk dietary habits (90%), but fewer had low-risk exercise habits (41%). Of 81 friends/relatives who participated, 25% had high-risk exercise habits. Student participants concurred the HIC is useful in developing lifestyle education programs. Challenges included uncertainty about obtaining laboratory data, serving-size quantities and confidence to effect lifestyle change in others. Although students appeared receptive to assessing health and lifestyle behaviors using the HIC, they reported being unconfident to prescribe long-term effective lifestyle advice. We recommend introducing the HIC in physical therapy curricula as an effective way of sensitizing emerging physical therapists to their responsibility to assess health/attributes/lifestyle non-communicable diseases (NCDs) risk factors. Prescribing lifestyle education/counselling warrants greater curricular focus. Further research will establish how HIC data and information can be effectively used as a clinical

FWTX-8005, to Xubo Wu. The funders had no role in study design, data collection and analysis, decision to publish, or preparation of the manuscript.

**Competing interests:** The authors have declared that no competing interests exist.

assessment and education tool to target health and lifestyle, and track behavior change over time.

## Introduction

Non-communicable diseases (NCDs) are a priority for the World Health Organization (WHO), particularly in countries with fast-growing economics such as China. The World Health Professions Association (WHPA) [1] including the World Confederation for Physical Therapy, one of the world's five leading health professions, has designed a Health Improvement Card (HIC or the Card) [2]. The HIC is a simple assessment and education tool to be used by health professionals to assess adults' health status in relation to common lifestyle-related attributes and NCD risk factors. The Card consists of two pages with four sections: patient biometric data; assessment of four modifiable risks factors (body mass index (BMI), fasting blood sugar and cholesterol level, and blood pressure (BP)); self-evaluation of lifestyle behaviors (diet, exercise, smoking and alcohol), and written commitment by the patient and the health professional to effect change in unhealthy biometric parameters or lifestyle behaviors. Level of health and/or disease risk is color coded as red (high risk), yellow (caution/moderate risk), and green (healthy or low risk) zones (S1 Appendix).

In their role as established health professionals, physical therapists are committed to maximizing each person's health, thus they have an essential role in health promotion and disease prevention [3–4]. To achieve this, contemporary physical therapy curricula need to prepare students accordingly [5–6]. To prepare students for their role in public health in China, a country with a fast-growing economy and significant escalation in NCDs [7], the concept of the HIC was introduced to second year students in two physical therapy programs in China by two of the investigators. To facilitate the application of the HIC in China, the WHPA HIC was translated into Chinese previously by one of the investigators based on the back translation method. This translation is now accessible on the WHPA website [1].

While the role of physical therapists in community and public health has garnered some attention in entry-level physical therapy education programs globally, albeit variable [8], whether physical therapy students are able to use the HIC after a brief standardized introduction to it, and whether they feel confident about using the HIC to assist their patients in lifestyle behavior change have not been examined. Thus, this study investigated whether a cohort of physical therapy students in China considered that their curriculum which included formal standardized introduction to the HIC and experience in administering it, provided a basis for assisting and providing lifestyle education and advice to others.

## Materials and methods

Approval to conduct this study was obtained from the Institutional Review Board of Shanghai University of Traditional Chinese Medicine and its affiliated hospital (Seventh People's Hospital of Shanghai University of Traditional Chinese Medicine) (Ethics approval number: 2018-IRBQY-013). The protocol for this study is available at dx.doi.org/10.17504/protocols.io. 54kg8uw.

### Sample

A purposive sample of volunteer physical therapy students was obtained from two participating WCPT-accredited physical therapy programs at universities in Shanghai, China. Students were assured anonymity and confidentiality of their responses to two questionnaires.

## Procedure

The experimental procedure had two parts. First, to personalize the experience of using the HIC, participating students were asked to complete the 2-page Chinese translation of the HIC (S2 Appendix). This was done after its standardized introduction in a standardized 45-minute classroom tutorial. The tutorial instruction was based on the Health Improvement Card User Guide for Health Professionals [9]. The tutorial, repeated at each of the two participating university programs, was given by the same instructor. Details of how to complete the Card were described. Height and weight of the student were self-reported; waist circumference and BP were measured using standardized procedures. Next, the student participants completed a questionnaire on their perceptions of the use and usefulness of the HIC. This questionnaire consisted of 7 statements and asked students to rate their level of agreement with each statement using a 4-point Likert scale (S3 Appendix). Second, to experience administering the HIC to another person, these same students invited a friend/relative to complete the HIC with the student serving in the role of the health professional. In a second questionnaire, they provided feedback on the strengths of use of the HIC and any challenges encountered in their role as a health professional, when assisting their friend/relative complete the Card. This questionnaire consisting of several open-ended questions asked students to a) identify aspects of the HIC and its administration that were effective in enabling a friend/relative to complete the HIC, b) describe any challenges encountered, and c) provide suggestions that could facilitate using and applying the HIC by them or others. Returned questionnaires were considered consent from students to participate.

## Data analysis

Descriptive statistics (frequencies and means±SD) were used to analyze the quantitative data, i.e., data reported in the HIC for the students and for the individual each interviewed, and for the responses to the first questionnaire based on Likert response options. Responses to the three open-ended questions in the second questionnaire distributed after the student interviewed a friend/relative were analyzed using established content theme analysis methods [10–12], and response frequencies.

The written responses from the student participants were typed and read multiple times by four of the investigators who were proficient in speaking and writing Chinese as well as English. Based on their consensus, multiple headings were used to describe various content themes [10–12]. The list of headings was then reviewed, regrouped, and coded and categorized into themes based on extraction of key words and phrases. The themes reported below emerged by consensus of the four specified investigators.

## Results

### Students' health status

The self-completed HICs were returned by 82 students (response rate = 100%). Participants were between 20–34 years of age. Mean BMI was 23.9±4.1 and 20.5±2.6 kg/m$^2$ for male and female students, respectively. Mean waist circumference for male students was 82.1±9.8 cm and 70.2± 7.7 cm for female students (Table 1, S4 Appendix). No participant had cholesterol or fasting blood sugar data. Over 80% of participants had low-risk (green code) BMIs (82%, n = 67) and BP (91%, n = 75). Approximately 90% (n = 73) of participants had dietary habits in the moderate-risk zone (yellow code); 41% (n = 34) exercised within the healthy low-risk zone (green code). Most with BMIs, BPs, diet and/or exercise habits in the moderate-risk

**Table 1. Quantitative data reported by students (n = 82).**

| Partici-pants | Number | Age (y) | Height (m) | Weight (kg) | BMI (kg/m²) | WC (cm) | BMI risk | BP risk | Action duration 1 | Diet risk | Exercise risk | Smoking risk | Alcohol risk | Action duration 2 |
|---|---|---|---|---|---|---|---|---|---|---|---|---|---|---|
| Men | 25 | 20–34 | 1.7± 0.06 | 71.1± 15.24 | 23.9± 4.07 | 82.1 ± 9.83 | G = 18 Y = 3 R = 4 | G = 19 Y = 3 R = 3 | 1 mon = 16 2 mon = 2 3 mon = 4 24 mon = 1 | G = 1 Y = 23 R = 1 | G = 16 Y = 9 R = 0 | G = 23 Y = 0 R = 2 | G = 25 | 1 mon = 18 2 mon = 2 3 mon = 3 4 mon = 1 2 y = 1 |
| Women | 57 | 20–34 | 1.6± 0.05 | 54.4± 8.27 | 20.5± 2.64 | 70.2 ± 7.67 | G = 49 Y = 3 R = 0 | G = 56 Y = 1 R = 0 | 1 mon = 40 2 mon = 6 3 mon = 4 4 mon = 6 6 mon = 1 | G = 7 Y = 50 R = 0 | G = 18 Y = 27 R = 12 | G = 57 | G = 57 | 1 mon = 40 2 mon = 6 3 mon = 2 4 mon = 6 5 mon = 2 6 mon = 1 |

BMI = body mass index, WC = waist circumference, BP = blood pressure, Action duration 1 = time committed to achieve BMI and BP to within low-risk zone, G = low-risk zone, Y = moderate-risk zone, R = high-risk zone, Action duration 2 = time committed to achieve life-style status to within low-risk zone. Data are mean±SD or number of students.

(yellow code) and high-risk (red code) zones were committed to achieving the low-risk (green code) zone within 1 to 2 months.

## Health status of students' friends/relatives

One student left the country to participate in an overseas exchange after initial completion of the HIC, therefore this student was unable to invite a friend/relative to complete the HIC and complete the second questionnaire about her experience with its application to another individual. Thus, a total of 81 participants completed HICs with another individual and returned them (response rate of 99%). Of the 81 friends/relatives who participated, most were in the 20–34 years old age group. In addition, most had low-risk BMIs (81%, n = 66) and BPs (79%, n = 64) (Table 2, S5 Appendix).

## Students' perceptions of the clinical use and application of the Card

All student participants agreed that physical therapists should be well familiar with the HIC and incorporate it into their patient assessments and lifestyle behavior change programs. Most (91%, n = 74) agreed that they understood the HIC's purpose and could identify instances of its use to improve patient outcomes (95%, n = 77) and provide lifestyle advice (96%, n = 78). Nearly 90% (n = 72) reported that they would recognize situations where the HIC might not be appropriate for a particular patient. However, 23% (n = 19) expressed some reservation about their ability to interpret the results effectively and/or progress a patient's health behavior change based on the HIC (Table 3, S6 Appendix).

## Feedback from students on assisting a friend/relative to complete the HIC

Two themes emerged from question 1 of the related questionnaire, three from question 2 and 4 themes from question 3 (Table 4).

**Aspects that worked well when assisting a friend/relative to complete the HIC.** Content analysis revealed that students' comments fell under two main themes. The first theme was the ease of use of the HIC. Sixty-six (81%) students commented on the straight forward data acquisition process for obtaining information such as age, height, weight, and BMI; and eating, smoking and exercise habits. The second theme was the ease of providing general advice to

**Table 2. Quantitative data reported by friends of the students (n = 81).**

| Partici-pants | Number | Age (y) | Height (m) | Weight (kg) | BMI (kg/m²) | WC (cm) | BMI risk | BP risk | Action duration 1 | Diet risk | Exercise risk | Smoking risk | Alcohol risk | Action duration 2 |
|---|---|---|---|---|---|---|---|---|---|---|---|---|---|---|
| Men | 29 | 20–34 = 25 35–39 = 1 50–54 = 1 | 1.8 ±0.42 | 69.7 ±11.93 | 22.6 ±63.7 | 80.6 ±10.46 | G = 22 Y = 5 R = 2 | G = 21 Y = 7 R = 1 | 1 mon-15 2 mon = 6 3 mon = 2 4 mon = 1 5 mon = 2 6 mon = 1 12 mon = 1 Life = 1 | G = 6 Y = 21 R = 2 | G = 14 Y = 8 R = 7 | G = 21 Y = 3 R = 5 | G = 23 Y = 4 R = 2 | 1 mon = 19 2 mon = 5 4 mon = 1 5 mon = 2 12 mon = 1 Life = 1 |
| Women | 52 | 20–34 = 44 35–39 = 2 40–44 = 2 50–54 = 4 | 1.6 ±0.06 | 56.0 ±9.66 | 21.0 ±3.16 | 70.8 ±8.8 | G = 44 Y = 6 R = 1 | G = 43 Y = 8 R = 1 | 1 mon = 36 2 mon = 4 4 mon = 5 5 mon = 1 6 mon = 3 7 mon = 1 9 mon = 1 13 mon = 1 | G = 10 Y = 42 R = 0 | G = 22 Y = 17 R = 13 | G = 49 Y = 1 R = 2 | G = 50 Y = 2 R = 0 | 1 mon = 34 2 mon = 5 3 mon = 1 4 mon = 3 5 mon = 2 6 mon = 1 7 mon = 1 9 mon = 2 12 mon = 1 13 mon = 1 Life = 1 |

BMI = body mass index, WC = waist circumference, BP = blood pressure, Action duration 1 = time committed to achieve BMI and BP to within low-risk zone, G = low-risk zone, Y = moderate-risk zone, R = high-risk zone, Action duration 2 = time committed to achieve life-style status to within low-risk zone. Data are mean±SD or number of students.

their friends/relatives on healthy lifestyle using the green (low-risk), yellow (moderate) and red (high-risk) zones to describe to their friends/relatives their risk levels of their health status.

**Challenges encountered when assisting friend/relative to use the HIC.** Three themes emerged regarding challenges encountered when assisting a friend/relative to complete the HIC. These themes were: 1) inability to obtain current blood cholesterol and fasting blood sugar levels, 2) lack of confidence in providing advice to effect lifestyle change in others which also included how to encourage friends/relatives to be persistent with a health improvement plan, and 3) uncertainty about the serving-size of vegetables and fruits in the nutrition section, and the volume of alcohol drinks.

**Table 3. Students' perceptions of the use and application of the Health Improvement Card.** Data are number (%).

| | Statements | Responses | | | | Total agreed (%) | Total disagreed (%) |
|---|---|---|---|---|---|---|---|
| | | Strongly Agree | Agree | Disagree | Strongly disagree | | |
| 1 | Physiotherapists should introduce the Health Improvement Card to the general public | 64 | 17 | 0 | 0 | 81 (100) | 0 (0) |
| 2 | I understand the purpose and role of the Health Improvement Card | 44 | 30 | 7 | 0 | 74 (91.4) | 7 (8.6) |
| 3 | I can provide advice to my patients about the actions prescribed on the Health Improvement Card | 44 | 34 | 3 | 0 | 78 (96.3) | 3 (3.7) |
| 4 | I can identify instances where using the Health Improvement Card would improve patient outcomes | 28 | 49 | 4 | 0 | 77 (95.1) | 4 (4.9) |
| 5 | I can justify my reasoning for choosing to implement the Health Improvement Card with my patients | 44 | 30 | 5 | 1 | 74 (91.4) | 6 (7.4) |
| 6 | I understand when using the Health Improvement Card may **not** be appropriate for a particular patient | 28 | 44 | 8 | 1 | 72 (88.9) | 9 (11.1) |
| 7 | I can interpret the results and/or progress a patient using the Health Improvement Card in an accurate manner | 16 | 45 | 18 | 1 | 61 (75.3) | 19 (23.5) |

**Table 4. Themes and number of responses to each open-ended question.**

| Themes | | Responses n (%) |
|---|---|---|
| Question 1: When assisting your friend/relative to complete the HIC, what are the aspects that work well? | | |
| 1 | Straight forward data acquisition | 66 (81%) |
| 2 | Provide general advice on healthy lifestyle | 13 (16%) |
| Question 2: What are the challenges encountered? | | |
| 1 | Difficulty to obtain blood cholesterol and sugar level | 36 (44%) |
| 2 | Confidence in provision of advice on persistency with healthy lifestyle | 36 (44%) |
| 3 | Uncertainty of serving-size of fruits or alcohol volume | 6 (7%) |
| Question 3: Can you think of anything that would help facilitate using and applying the HIC? | | |
| 1 | Hospital involvement–educational courses (online or face to face) / HIC to be administered in hospital departments/ HIC to be used in conjunction with regular health check | 25 (31%) |
| 2 | Inclusion of sleep-health and mood-related questions in the HIC | 9 (11%) |
| 3 | Provide detailed information on serving size of fruits and volume of '1 drink' of alcohol | 6 (7%) |
| 4 | Availability of portable equipment for physiotherapists to obtain cholesterol and sugar data | 5 (6%) |

**Suggestions to facilitate HIC use by health professionals.** Almost one third of the students (n = 25) suggested that public education is required to introduce the concepts of the HIC, for example via online courses, and that such an initiative could be led by hospital departments. Further, they recommended that the HIC be used by health professionals, including physical therapists, in conjunction with patients' annual health check-ups. Nine (11%) students suggested that more health-related questions such as sleep health and mental health and mood-related information should be included in the HIC. Other suggestions included physical therapists' use of portable equipment to obtain accurate and current cholesterol and fasting blood sugar data, and the need to clearly operationalize dietary and alcohol quantities vis-à-vis serving sizes.

## Discussion

Based on the WHO's strategic action plans [13–14] and United Nations high-level meetings on NCDs [15], much needs to be done at the primary health care level as well as policy level to address contemporary issues associated with NCDs globally. This has become a particular priority in low- and middle-income countries such as China, where morbidity, disability, and mortality related to NCDs are escalating at disproportionate rates [7]. The practice of physical therapy particularly at an international standard is relatively new to China [16], thus physical therapy leaders need to focus on tailoring the profession to best meet the needs of Chinese society. This is the first study to describe a health profession's curriculum, in particular physical therapy curriculum, that incorporates standardized instruction on the established and WCPT-supported HIC, developed by the WHPA for use by these health professionals. The instruction was based on a companion user manual also published by the WHPA. In addition, this is the first study to examine its use by students, who are tomorrow's physical therapists, with respect to their perceptions about its strengths and ease of use and its usefulness, and their perceived competence in prescribing health behavior change programs for others.

Few studies have examined health sciences students' perceptions of their personal health and their capacity to maximize outcomes as qualified health professionals. Research conducted in the Middle East has shown marked discrepancy between health knowledge, and beliefs and practices in staff as well as students on a health sciences campus in Kuwait [17]. The findings underpinned an initiative to promote health on campus in health sciences students and staff,

as the literature supports healthy health professionals are more likely to support healthy life-style in their patients and their families, and their advice is viewed more credibly by patients [18–20].

The findings of the present study supported a positive attitude of physical therapy students toward a health and lifestyle behavior assessment/evaluation and education tool, namely, the HIC. They appeared to appreciate the need to address global health indices and lifestyle behaviors in every patient, and that this should extend to all physical therapists. They identified the tool as easy to administer and facilitated engaging a person to discuss their health and lifestyle behaviors and attributes that may be injurious to them. In addition, it was thought to provide a basis for developing personalized lifestyle programs, even though the students were not confident as yet about effecting behavior long-term change in others.

After self-administrating the Card and then administering to one other individual, the students shared relevant insights about how the Card and its administration by physical therapists could be improved. The investigators largely concurred with the students. First, recent cholesterol and fasting blood sugar levels may not be readily available. These biometric data are fundamental indices of NCD risks, as well as the basis for individualized advice on lifestyle modification. Physical therapists need to be able to request these laboratory tests or have portable devices for such measurements. Second, while acceptable levels of daily alcohol consumption are available in the HIC health professional user guidelines, definition of serving sizes or portions of vegetables and fruit are unclear. In addition, sleep hygiene and mental health warrant inclusion into the next iteration of the HIC. The investigators are forwarding these recommendations to the WHPA for their consideration for inclusion into the next iteration of the HIC and the user guidelines.

Our findings supported that Chinese physical therapy students lacked confidence in effecting long term behavior change. This is consistent with that reported for qualified physical therapists vis-à-vis smoking cessation, nutritional advice, and recommendations about being more physically active [21–24]. Students need to be taught self-efficacy competences associated with health and lifestyle assessment as a basis for designing lifestyle behavior change programs, and implementing them effectively. Recently, Dean and colleagues published health competency standards in physical therapy practice which promotes routine inclusion and use of the HIC [4].

A potential limitation of the study is that height and weight of our students were self-reported, thus the validity of the calculated BMI may be questionable. However, all university students in China are required to participate in compulsory physical education classes and must have their height and weight recorded at least once a year. Students were expected to report these official data, therefore error in BMI should be minimal.

This pilot study first aimed to explore whether entry-level physical therapy curriculum at two physical therapy programs in Shanghai adequately prepared the students to assess health and NCD risk factors and, on that basis, enable them to provide health and lifestyle advice. Our findings have provided a foundation for designing and launching a community-based study, with physical therapy students (supervised by their clinical educators) administering the HIC to attendees at a community center (n = 80). Based on individual data from the HIC, students engage participants in designing health promotion programs and evaluating the outcome over time.

Finally, further studies are needed to replicate and extend these findings to other countries and cultural contexts. The HIC has been translated into multiple languages, thus large-scale multicultural studies can be implemented readily with practitioners as well as students.

## Conclusion

Our findings support that introducing the HIC early in physical therapy curricula, even in a single short tutorial coupled with practice, may enhance students' confidence in assessing health and NCD lifestyle risk factors. Although students largely agreed that information obtained from the HIC can assist them in targeting health education to a given individual, they expressed less confidence in prescribing effective long-term lifestyle behavior change to assist people in reducing their NCD risk factors. Overall, our findings support the need for dedicated attention to health and lifestyle assessment and related counselling in physical therapy curricula. This finding lends support for the inclusion of health competencies for effective health and lifestyle behavior change. In summary, the HIC is a promising clinical tool for use by students, for readily assessing health and lifestyle behaviors, and, on such basis, prescribing effective lifestyle behavior change programs. The HIC has important potential in enabling patients to support their health and wellbeing and reducing risk of NCDs that are of epidemic proportions globally. Further, its data can serve as a basis for follow-up and tracking progress of health status and lifestyle change over time. Further studies are needed to promote the uptake and use of the HIC by clinicians, educators and students across settings and countries, consistent with the mandate and values of WCPT.

## Supporting information

**S1 Appendix. English version of the Health Improvement Card (Reprinted with permission).**
(DOCX)

**S2 Appendix. Chinese version of the Health Improvement Card (Reprinted with permission).**
(DOCX)

**S3 Appendix. Students' perceptions of the use and application of the Health Improvement Card.**
(DOCX)

**S4 Appendix. Data reported by individual student.**
(PDF)

**S5 Appendix. Data reported by each student's friend/relative.**
(PDF)

**S6 Appendix. Individual student's responses to the perception questionnaire.**
(PDF)

## Acknowledgments

We gratefully acknowledge the students and their friends or relatives who participated in this study. This project was partially funded by the Shanghai Municipal Health Bureau, project number: ZY (2018–2020)—FWTX-8005.

## Author Contributions

**Conceptualization:** Xubo Wu, Alice YM Jones, Yiwen Bai, Jia Han, Elizabeth Dean.

**Data curation:** Xubo Wu, Yiwen Bai, Jia Han.

**Formal analysis:** Xubo Wu, Alice YM Jones, Yiwen Bai, Jia Han, Elizabeth Dean.

**Funding acquisition:** Xubo Wu.

**Investigation:** Xubo Wu, Alice YM Jones, Yiwen Bai, Jia Han, Elizabeth Dean.

**Methodology:** Xubo Wu, Alice YM Jones, Yiwen Bai, Jia Han, Elizabeth Dean.

**Project administration:** Xubo Wu, Alice YM Jones, Yiwen Bai, Jia Han, Elizabeth Dean.

**Resources:** Alice YM Jones, Elizabeth Dean.

**Supervision:** Xubo Wu, Alice YM Jones, Yiwen Bai, Jia Han, Elizabeth Dean.

**Validation:** Xubo Wu, Alice YM Jones, Yiwen Bai, Jia Han, Elizabeth Dean.

**Writing – original draft:** Alice YM Jones.

**Writing – review & editing:** Xubo Wu, Alice YM Jones, Yiwen Bai, Jia Han, Elizabeth Dean.

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
