## [Decision Letter · Decision Letter 0]

2 Aug 2019

PONE-D-19-20237

Health Improvement Card application in Chinese physical therapy students - A pilot study

PLOS ONE

Dear Professor Jones,

Thank you for submitting your manuscript to PLOS ONE. After careful consideration, we feel that it has merit but does not fully meet PLOS ONE’s publication criteria as it currently stands. Therefore, we invite you to submit a revised version of the manuscript that addresses the points raised during the review process.

We would appreciate receiving your revised manuscript by Sep 16 2019 11:59PM. To enhance the reproducibility of your results, we recommend that if applicable you deposit your laboratory protocols in protocols.io, where a protocol can be assigned its own identifier (DOI) such that it can be cited independently in the future. For instructions see: http://journals.plos.org/plosone/s/submission-guidelines#loc-laboratory-protocols

We look forward to receiving your revised manuscript.

Kind regards,

Shane Patman, PhD

Academic Editor

PLOS ONE

Journal Requirements:

2. Please update the ethics statement on the submission form and the methods section of your manuscript file to remove the word "Blinded" and replace with the named IRB that approved your study.

Reviewers' comments:

Reviewer's Responses to Questions

**Comments to the Author**

1. Is the manuscript technically sound, and do the data support the conclusions?

Reviewer #1: Yes

Reviewer #2: Yes

2. Has the statistical analysis been performed appropriately and rigorously? 

Reviewer #1: Yes

Reviewer #2: Yes

3. Have the authors made all data underlying the findings in their manuscript fully available?

Reviewer #1: Yes

Reviewer #2: Yes

4. Is the manuscript presented in an intelligible fashion and written in standard English?

Reviewer #1: Yes

Reviewer #2: Yes

5. Review Comments to the Author

Reviewer #1: I enjoyed reading this well-written manuscript and feel that it adds some value to the existing knowledge in physiotherapy education research.

Some comments for the authors to consider:

a) Please indicate n= in the text of Results - currently only percentages are presented.

b) Table 1 - there are no data presented for males under the column labelled 'Action duration 2' - please revise and add.

c) The use of abbreviations introduced into the text is not consistent throughout the different sections of the manuscript - please revise.

d) Grammatical errors:

Line 205 - should be 'professionals'

Line 222 - should be 'therapists'

Lines 230 - 236 - please revise sentence construction and language use

e) Conclusion: consider re-writing in a more concise manner.

Reviewer #2: Please find below some general comments for consideration. There are also a few minor grammatical errors to note.

The title indicates this is a pilot study; as such the feasibility and next steps in moving to a bigger study are not fully considered in the discussion (there is a line about on time/burden which is helpful).

Conclusion in the Abstract (Line 22) and the main conclusion (line 257) – it is not quite clear what the term re-evaluation means. Is it responsiveness i.e. whether the HIC is responsive to change following an intervention?

Introduction

line 46 ‘receptive’ vs 'receptive to' or 'able to'

Lines 51-54 appear to include the study aims but these seem a bit different to the start of the abstract; consistency and explicit identification of the study aims would be helpful.

Methods

Line 75 “height and weight were self-reported” Perhaps this should be considered/discussed as a potential limitation to the study, given that frequently there are inconsistencies between self-report and accurate measurement. Physiotherapy students should be able to measure height and weight with minimal equipment/expense but perhaps I am assuming this for all undergraduate programmes.

Line 85 “instruction provided identify” Vs “instruction provided to identify”

Results

Line 124 “she” vs “they were”

Discussion

Line 191 “….reverse the tide of ….” Please consider use of more scientific language here

Line 209 “given” Vs “as”

224-225 – This is not quick clear. What does “order” mean; does it mean to request these results from a medical site?

Line 230 “That the students….” Rephrase the start of this sentence please

233 – Does this mean the authors will now contact the WHPA with these results, it may be worth considering whether a recommendation can be made at this stage based on a pilot study.

6. PLOS authors have the option to publish the peer review history of their article (what does this mean?). If published, this will include your full peer review and any attached files.

Reviewer #1: No

Reviewer #2: No

---

## [Author Response · Author response to Decision Letter 0]

8 Aug 2019

We are grateful to the editor and reviewers for their comments and suggestions and have revised the manuscript accordingly. Please refer to the file "Responses to Reviewers" for our detailed responses to each reviewer's comments.

---

## [Decision Letter · Decision Letter 1]

13 Aug 2019

Use of Health Improvement Card by Chinese physical therapy students: A pilot study

PONE-D-19-20237R1

Dear Dr. Jones,

We are pleased to inform you that your manuscript has been judged scientifically suitable for publication and will be formally accepted for publication once it complies with all outstanding technical requirements.

With kind regards,

Shane Patman, PhD

Academic Editor

PLOS ONE

Additional Editor Comments (optional):

Reviewers' comments:

Reviewer's Responses to Questions

**Comments to the Author**

1. If the authors have adequately addressed your comments raised in a previous round of review and you feel that this manuscript is now acceptable for publication, you may indicate that here to bypass the “Comments to the Author” section, enter your conflict of interest statement in the “Confidential to Editor” section, and submit your "Accept" recommendation.

Reviewer #2: All comments have been addressed

2. Is the manuscript technically sound, and do the data support the conclusions?

Reviewer #2: (No Response)

3. Has the statistical analysis been performed appropriately and rigorously? 

Reviewer #2: (No Response)

4. Have the authors made all data underlying the findings in their manuscript fully available?

Reviewer #2: (No Response)

5. Is the manuscript presented in an intelligible fashion and written in standard English?

Reviewer #2: (No Response)

6. Review Comments to the Author

Reviewer #2: (No Response)

7. PLOS authors have the option to publish the peer review history of their article (what does this mean?). If published, this will include your full peer review and any attached files.

Reviewer #2: No

---

## [Editor Report · Acceptance letter]

28 Aug 2019

PONE-D-19-20237R1 

Use of the Health Improvement Card by Chinese physical therapy students: A pilot study

Dear Dr. Jones:

I am pleased to inform you that your manuscript has been deemed suitable for publication in PLOS ONE. Congratulations! Your manuscript is now with our production department. 

With kind regards,

on behalf of

Assoc Prof Shane Patman 

Academic Editor

PLOS ONE